# Enhancing Maritime Cybersecurity through Operational Technology Sensor Data Fusion: A Comprehensive Survey and Analysis

**DOI:** 10.3390/s24113458

**Published:** 2024-05-27

**Authors:** Georgios Potamos, Eliana Stavrou, Stavros Stavrou

**Affiliations:** Faculty of Pure and Applied Sciences, Open University of Cyprus, Latsia, 2231 Nicosia, Cyprus; eliana.stavrou@ouc.ac.cy (E.S.); stavros.stavrou@ouc.ac.cy (S.S.)

**Keywords:** maritime cybersecurity, maritime OT data fusion, maritime cyberthreat detection

## Abstract

Cybersecurity is becoming an increasingly important aspect in ensuring maritime data protection and operational continuity. Ships, ports, surveillance and navigation systems, industrial technology, cargo, and logistics systems all contribute to a complex maritime environment with a significant cyberattack surface. To that aim, a wide range of cyberattacks in the maritime domain are possible, with the potential to infect vulnerable information and communication systems, compromising safety and security. The use of navigation and surveillance systems, which are considered as part of the maritime OT sensors, can improve maritime cyber situational awareness. This survey critically investigates whether the fusion of OT data, which are used to provide maritime situational awareness, may also improve the ability to detect cyberincidents in real time or near-real time. It includes a thorough analysis of the relevant literature, emphasizing RF but also other sensors, and data fusion approaches that can help improve maritime cybersecurity.

## 1. Introduction

Cybertechnologies have become key factors for the operational management of numerous systems, as well as the safety and security of critical sectors such as the maritime industry [1]. Information technology (IT) and operational technology (OT) systems, either maritime or industrial, are installed on board ships or in ashore maritime infrastructures or can be interlinked between ship and shore extending the internet or intranet links to the oceans [2]. Furthermore, the digitization era has resulted in the emergence of autonomous/unmanned ships, which has created new requirements for interoperability and secure connectivity [3,4].

The use of satellite links has significantly improved connectivity, allowing for efficient ship-to-shore data exchange. Satellite and radio data lines provide the necessary bandwidth to facilitate real-time data exchange through remote access [5]. This advancement is essential for efficient communication, navigation, and overall operational efficiency [6]. However, the increased connectivity and integration of sensors/actuators into command-and-control (C2) systems or human–machine interfaces (HMIs) also expand the potential cyberattack surface. The enlarged attack surface poses a threat as attackers may exploit vulnerabilities in the systems, leading to malicious acts. The risks associated with cyberthreats in the maritime industry include unauthorized access, data breaches, system manipulation, or even the compromise of autonomous systems.

According to [2], cyberattacks can be either targeted or untargeted, with potential targets encompassing both IT and OT systems. Specifically, in targeted OT systems, vulnerabilities may span a range of maritime-specific OT systems utilized for navigation, surveillance, information exchange, vessel management, cargo scheduling, crew and passenger management, entertainment, and safety [2,7,8]. Threats such as malware, denial of service (DoS) attacks, spoofing, supply chain attacks, and brute force attacks exploit these vulnerabilities, guided by threat models like MITRE ATT&CK [9] and Lockheed Martin [10]. Consequently, attack detection plays a pivotal role in cyber risk management, crucial for the effective mitigation of potential impacts.

Detecting cyberthreats or anomalies in the complex and technology-rich maritime environment presents a challenging task. When incidents affect navigational or surveillance systems, it is important to determine their causes, which could range from environmental factors and system malfunctions to maneuvers or kinematic operations. Therefore, command and control, or information and event management systems, must adeptly identify whether an incident stems from cyberexploitation of a vulnerable maritime asset/system or represents a non-cyber-related maritime anomaly. These systems enhance maritime situational awareness (MSA) by integrating and analyzing data from surveillance sensors and other sources, enabling the detection of maritime anomalies.

Current research on real-time detection of cyberattacks in the maritime sector is limited [11,12], with prior studies focusing mainly on leveraging sensors or similar technologies to identify threats aimed at information and communications technology (ICT) elements within operational technology (OT) systems, especially on board ships. This survey discusses detection methods for cyberattacks targeting maritime OT systems, utilizing data not just from IT equipment (such as logs, alerts, events, files, etc.) but also from navigational and surveillance systems that employ protocols for localization, target detection, and maneuvering, many of which are known to be vulnerable [2,13,14]. Understanding the maritime cyberthreat landscape, including the systems and protocols at risk, as well as detection technologies and methodologies, is essential for developing effective cyberdetection capabilities to safeguard against maritime cybersecurity incidents.

The development of mechanisms for cyberattack detection in the maritime domain necessitates a detailed analysis of maritime data, facilitated by continuous monitoring of navigational and surveillance sensors. Utilizing operational data from surveillance and navigational sensors to enhance cybersecurity in the maritime sector represents a promising research avenue. The scope of this paper is to identify the radio frequency (RF) sensors that can be utilized for the composition of the maritime situational awareness (MSA) and then investigate their potential for maritime cyber situational awareness (MCSA). The paper examines if data fusion of such sensors used for the MSA establishment can also assist the detection and response to cyberthreats within the maritime domain, thereby contributing to the overall security and resilience of maritime operations.

Section 2 presents the methodology followed in this paper, and Section 3 summarizes and presents the relevant maritime cyberthreat landscape. At this point, radar and AIS technologies are described, and they appear to be common elements that can be combined to form the maritime picture. Section 4 investigates whether data fusion may improve the detection of cyberthreats, to be potentially utilized for both maritime and cyber situational awareness. Section 5 discusses future directions, and Section 6 summarizes the findings.

## 2. Methodology

To support the cybersecurity objectives of this research in the maritime domain, a three-step methodological approach was employed, as illustrated in Figure 1. The initial step involved the identification of the maritime domain’s attack surface and its cyberthreat landscape, informed by the legislative framework, current regulations, guidelines, and relevant maritime cybersecurity literature. This encompassed sensors and systems, potential interconnections among them, protocols, and networks, collectively constituting the attack surface. Subsequently, an analysis and taxonomy of documented cyberincidents against navigational and surveillance networks and systems offered insights into vulnerabilities and the potential repercussions of their exploitation.

This analysis was expected to pinpoint cybervulnerabilities in surveillance and navigational systems such as the automatic identification system (AIS), radar, Global Navigation Satellite System (GNSS), and electronic chart display and information system (ECDIS) employed both on board ships and within ashore maritime infrastructures. The International Convention for the Safety of Life at Sea (SOLAS) sets essential safety standards for the construction, equipment, and operation of merchant ships, mandating the use of radar, GNSS, and AIS for larger vessels. Ref. [15] highlights the operational and technical features of surveillance sensors, the integration of radar and AIS data, and the architecture of an integrated bridge system (IBS).

The second phase of this work focused on identifying and cataloging studies related to the data fusion of radar and AIS, to investigate their potential to be used for detecting cyberthreats. The concept of AIS and radar data fusion was first introduced in 2008 [16], marking a significant development in maritime security technology. Ref. [17] discusses an attack on compromised electronic chart display and information systems (ECDIS), highlighting vulnerabilities in command-and-control systems that integrate data from radar and AIS. This paper is particularly interested in the application of data fusion for detecting cyberincidents alongside the formation of the Recognized Maritime Picture (RMP), emphasizing the dual capability of identifying maritime anomalies and cyberincidents through the integration of AIS and radar data. In the maritime context, “anomaly” encompasses any abnormal ship behavior due to environmental conditions, system malfunctions, or maneuver/kinematic operations, underscoring the broad scope of this investigation. Therefore, the key factors taken into consideration when examining AIS and radar data fusion are the following:AIS and radar have different RF characteristics [15], and their combined operation and data fusion provide essential benefits for detecting cyberincidents, especially related to AIS vulnerabilities. Radar is used only for detection purposes, while AIS is commonly used for identification.Bridge systems contain a variety of vulnerabilities that, if exploited, can impact the reliability/accuracy of the MSA, thus impacting both the safety and security of ships. Due to this, a focused literature survey was performed in the context of cybersecurity and anomaly detection to identify papers that identify anomalies related to the operation of the bridge systems. Most of these anomalies may not be referred to as cyberincidents but may be connected to cyber risks and as such have been considered in this study.Existing investigations indicate that there are cases where AIS anomalies are caused by falsified transmitted messages/data. For this reason, it was necessary to use additional search keywords during the focused literature survey referring to the transmitted AIS signal and/or the “angle of arrival” to the receiver.

To identify the relevant research work, the following keywords were considered: (a) AIS and radar “data fusion”, which resulted in 1630 research manuscripts; (b) “data fusion” for maritime cybersecurity, which resulted in 133 manuscripts; and (c) AIS signal and “angle of arrival” (exact phrase anywhere in the article), which resulted in 363 manuscripts.

Overall, 2126 manuscripts matched the search criteria. Then, a set of selection criteria have been applied to pinpoint the manuscripts of interest, including the ones retrieved by several researchers in various systematic reviews [18,19]. A manuscript was selected if it was an article, it was published in an international peer-reviewed journal or conference, it was written in English, and it reported empirical data. Technical reports, book chapters, abstracts, and literature reviews considered not relevant were excluded. A manuscript was also excluded if data fusion was not performed by sensors on board a ship and/or in a maritime infrastructure. According to this process, 227 papers were selected for further process.

The manuscripts were reviewed in detail, looking for patterns of common interest to this work. The analysis was performed taking into consideration the most important elements of data fusion according to [20]: data sources, their operation based on the applied algorithm, and the purpose of the solution. The process was repeated for each topic separately, illustrating the connection/combination of the various tasks/areas.

## 3. Initial Phase—Cyberthreat Landscape in the Maritime Context

This section examines the cyberthreat landscape in the maritime environment, to provide the necessary insights about if maritime OT (surveillance and navigation) data fusion can be used for detecting cyberthreats. 

To following paragraphs in this section are defined in accordance with the NIST CFW actions, which are related to the cyberthreat landscape. Figure 2 presents the National Institute of Standards and Technology (NIST) Cybersecurity Framework (CFW) and the actions related to the development of cyberthreat detection capabilities. The identification includes the related legal and regulatory framework for maritime assets, mission objectives, and an assessment of operational criticality. To describe the maritime operations and mission objectives, we present the cyberincidents that involved navigational and surveillance systems, followed by a taxonomy of the infected systems and attack types. 

### 3.1. Legal and Regulatory Framework

Understanding the relative guidelines and responsibilities is a critical step in defining the cyberthreat landscape in the maritime domain. The goal is to create a comprehensive regulatory and legislative framework that promotes cyber risk identification and management. BIMCO, in cooperation with many shipowners [2], published guidelines to manage risks on board ships, considering the NIST Cybersecurity Framework. The International Maritime Organization (IMO) adopted resolution MSC 428 (98), to address the cyber risks in existing management systems. Additionally, IMO recommended in a circular, MSC-FAL.1/Circ.3, high-level measures to safeguard ships from cyberthreats and vulnerabilities. ENISA published guidelines for cyber risk management for ports [21]. The Advance Bureau of Shipping published guidelines for the cyber resilience of ships (IACS UR E-26) and of onboard systems and equipment (IACS UR E-27). The Digital Container Shipping Association analyzed the maritime cyber risks, based on the NIST Framework. The International Association for Classification Societies has issued a “Recommendation on Cyber Resilience (No. 166)” [22]. The Maritime Bulk Liquids Transfer Cybersecurity Framework Profile assists in cybersecurity risk assessments for all involved entities [23]. All the above guidelines designate the relevant cybersecurity requirements. 

### 3.2. Maritime Assets 

The maritime sector is undergoing a profound transformation due to technological advancements, evolving into a realm characterized by digital connectivity, intelligence, and autonomy. Legacy systems and protocols have been integrated into the Internet Protocol (IP), unlocking numerous networking opportunities previously unattainable. However, this digital shift renders ships and critical maritime infrastructures—such as ports, energy transport systems, supply chain facilities, and operational centers belonging to either government or private entities—increasingly susceptible to cyberthreats. Consequently, the maritime industry is compelled to devise strategies and enhance its capabilities to safeguard maritime assets against cyberattacks. This involves protecting a diverse array of systems and sensors, each employing different technologies ranging from informational to navigational and surveillance functions.

As shown in Figure 3, the environment includes mobile platforms, such as ships and autonomous vessels, as well as coastal infrastructures. The majority of coastal facilities are classified as critical infrastructure, e.g., ports, energy storage facilities, water facilities, etc. Systems on board ships and vessels can operate independently or in conjunction with ashore command and control systems, exchanging data and information. All platforms are capable of internal and external digital communication via RF links or satellite communications (SATCOM). Cargo, tanker, passenger, fishing, and cruise ships; autonomous surface vessels (ASVs); and other automated vehicles, oil rigs, and other supporting vessels are examples of onboard means. Similarly, maritime infrastructures include governmental or private sector maritime operational centers, observation/surveillance posts, and critical infrastructures such as ports or energy transportation facilities. Ships and infrastructures have distinct communication and operational capabilities that must be identified in order to effectively develop cyber detection tools and mechanisms.

To develop cyberthreat detection capabilities effectively, it is essential to first gain a comprehensive understanding of the maritime domain, pinpointing all components requiring protection and identifying the cyberthreats that pose risks to maritime operations. Additionally, before exploring the technical details of detecting cyberattacks, it is crucial to analyze the maritime domain through the lenses of complexity and criticality. 

The complexity of the maritime domain makes the successful detection of cyberthreats difficult because it necessitates a clear view of all relevant tasks that must be completed. Measures for detecting malicious activity include immediate sensing and alerting. In a previous work [24], relevant tasks for effective detection were identified and listed across the NIST Cybersecurity Framework. Given that these tasks are listed in different functional areas, it is frequently unclear which of these are strongly interconnected and can impact detection effectiveness.

The criticality of the maritime domain requires real-time or near-real-time capabilities to support the continuous determination of the situation, relative to the maritime domain’s mission. Given that IT/OT operations support maritime tasks, the cybersecurity domain is interlinked with the maritime domain and contributes to the MCSA. Cybersafety concerns the risks from the loss of availability, integrity of the safety critical data, and reliability of the OT systems [2]. Both the security and safety of this critical domain necessitate taking the necessary steps to reduce the cyber risk and the severity of the potential impact.

### 3.3. Maritime Operations and Mission Objectives

The exposure of the maritime domain to a range of cyberthreats forced the maritime industry to develop procedures and capabilities to protect its mission objectives [13], effectively maintaining preparedness, personnel and environmental safety, and operational security. This section contains an analysis of the cyberincidents against the maritime OT equipment and, for each incident, the type of attack. 

#### 3.3.1. Analysis of Cyberincidents That Involved Navigational and Surveillance Systems

Reported maritime incidents/events at sea caused severe disasters, damages, and human losses or affected important work [25]. These incidents were taxonomized and categorized in ref. [26], using several criteria such as the entry point and the infected system. In this paper, the work in ref. [13] is discussed to provide a clear overview of an attacker’s capabilities, concerning the vulnerabilities’ exploitation of the navigational and surveillance systems, that may change the security and safety situation in this complex environment [27]. Table 1 presents a list of recent cyberincidents in the maritime domain, identifying the attacks that were executed against specific sensors/systems, along with details regarding the infected systems and the type of attack. This analysis provides a first indication of the systems utilized in MSA that can constitute a cybertarget and identify the potential impact. 

#### 3.3.2. Infected Systems

As listed in Table 1, systems that can be affected by a cyberattack include navigational and surveillance systems, such as ECDIS and GNSS. Further investigations confirmed the systems under attack as depicted in Table 1 and provided more insights as to the vulnerable systems that exist on ships and onshore maritime infrastructures/centers [2,7,8,25,34,35,36]. As Figure 4 highlights, vulnerable systems include sensors, communication or management/control systems.

#### 3.3.3. Types of Attacks

To further analyze the cyberthreat landscape, the authors used the MITRE ATT&CK framework [24], utilized to develop a blueprint for detecting cyberthreats. The purpose of the proposed blueprint was to provide insights into the behavior and techniques that hackers use to exploit the vulnerabilities of maritime OT, navigation and surveillance, sensors, and actuators. The cyberattack kill chain is expected to guide the security team to enhance systems’ detection capabilities. Especially for this category of sensors/actuators, attacks can be categorized as follows:Malware is malicious software that is designed to access or damage a computer system without the knowledge of the owner. There are various types of malware including trojans, ransomware, spyware, viruses, and worms. Ransomware encrypts data on a system until a ransom has been paid. It can be used for denial-of-service purposes. Malware may also exploit known deficiencies and problems in outdated/unpatched software.Brute force techniques try to guess the credentials of a network device through repeated attempts.Denial of service (DoS) techniques prevent authorized users from accessing information, usually by flooding network devices (computers and servers) with data. DoS is also applicable to OT systems.Man-in-the-middle attempts acting as a form of active eavesdropping attack, in which the attacker intercepts to read or modify data communications to masquerade as one or more of the ship’s entities involved.Spoofing attacks where a false signal is broadcasted with the intent to mislead the victim receiver, such as AIS or a Global Positioning System.Sophisticated attacks are conducted on the navigation network and surveillance systems since many of them are integrated into shoreside networks for updating and provision of services purposes.Supply chain attacks.The analysis of the existing cyberincidents clearly suggests the potential maritime systems that can be targeted. A maritime cybersituational awareness (MCSA) picture has to be established for effective cyberthreat detection against vulnerable navigational and surveillance systems. Subsequent sections will examine if data fusion from the relevant maritime sensors can enhance MCSA.

## 4. Findings of the Literature Review Related to the Maritime Data Fusion

The effective identification of the attack landscape in the maritime domain provides insights into potential cyberthreats. The knowledge gained can provide initial direction to cyberincidents, as well as drive threat detection and further analysis. Because the term “maritime data” can have a wide range of interpretations, it was necessary to define what these data entail. As such, the analysis of the threat landscape was used to clarify which systems/sensors are used for safe navigation and effective surveillance and to identify their vulnerabilities. Mandatory maritime sensors include radar and AIS, according to the SOLAS convention [37]. Therefore, to identify the relevant papers for maritime data fusion, the search criterion “radar and AIS fusion” was mainly utilized to discover the relations between radar and AIS in the context of data fusion.

The findings of this extensive literature review were used to investigate the application of data fusion and to identify the data that should be collected. These data could point the way toward establishing both a reliable MSA and an effective real-/near-real-time cyberincident detection mechanism. Therefore, as mentioned in the Section 2 above, the analysis was performed according to [20], considering the key elements of data fusion, including purposes, data sources, and operations. 

### 4.1. Purpose 

“Radar and AIS fusion” can improve situational awareness by minimizing errors introduced by specific sensor accuracy. Furthermore, data correlation provides additional intelligence capable of improving the security and safety of maritime means. The authors of this work discovered and mapped, in Table 2, ten categories related to the purpose of data fusion based on an examination of the collected material. Figure 5 shows an illustration of the map. These ten main categories were divided into three groups: target tracking, intelligence, and security and safety.

Focusing on security and safety, AIS/radar data fusion could ensure safety on board, safe navigation, and accurate positioning. Safe navigation and accurate positioning are some of the most essential factors for achieving secure behavior at sea. In the same manner, the localization accuracy [38] and collision avoidance are also examined, along with the tracking on the ECDIS view [39]. 

For the detection of AIS anomalies, real-/near-real-time and historical data are used. As for the real-/near real-time data process, AIS spoofing is classified as an incident that could be related to malicious cyberactivities. The detection of AIS spoofing is examined in [40,41,42]. Maritime mobile service identity (MMSI) spoofing is examined in [43]. Also, the transmission of erroneous AIS messages could be realized as in [44], and the AIS messages’ falsification can be detected as in refs. [45,46]. In addition to that, the detection of the intended AIS on/off switch is discussed in ref. [47]. 

The exploration of historical AIS data has been used mostly to improve the accuracy of vessel positioning information. This analysis can be also part of a malicious cyberdetection mechanism. As with the algorithmic process of historical AIS data, it is possible to perform self-reporting, vessel trajectory reconstruction [48], or learning [49] or to detect suspicious vessel activities [50]. Historical data could be also used to understand the behavior of the ship [51,52]. In addition to the non-real-time analysis, AIS data have been studied in the context of maritime image processing [53].

As already mentioned, the maritime domain contains anomalies, which are not AIS related. An approach presented in ref. [54] demonstrates how an implementation using the ELK stack (Elasticsearch, Logstash, Kibana) can be used for detecting maritime anomalies. The strange/suspicious kinematic behavior of a ship, the dangerous maneuvers, malfunctions of industrial (including marine engines) or electronic equipment, and security incidents are typical anomalies. According to the findings, data fusion has been utilized for security purposes, for both the detection and analysis of maritime anomalies including AIS. 

From the cybersecurity perspective, the application of data fusion for the detection of maritime cyberincidents is partially examined. The ISOLA project enforces mechanisms to detect, among others, cyberincidents impacting the information technology on board passenger ships [55]. In addition to the existing literature, this paper describes an algorithm that uses maritime data fusion of navigational and surveillance sensors, contributing to the existing research background for the following purposes:It specifies the sensors used (radar, AIS, and direction finding (DF)).It is capable of cyberincidents’ real-time detection related to maritime OT systems.It can be used on board and ashore.

**Table 2 sensors-24-03458-t002:** Purposes of AIS/radar data fusion.

A/A	Purpose of AIS/Radar Data Fusion	Representative Publications
1.	Improve miss the target/false alarms/noisy observations	[56,57,58,59,60,61,62,63,64,65]
2.	Ensure safety/navigation/positioning	[38,39,66,67,68,69,70,71,72,73,74,75,76,77,78,79,80,81,82,83,84,85,86,87,88,89,90,91,92,93,94,95,96,97]
3.	Solve track association problem	[15,59,67,68,98,99,100,101,102,103,104,105,106,107]
4.	Improve reliability and accuracy of tracking	[59,69,80,103,108,109,110,111,112,113,114,115,116,117,118,119,120,121,122,123,124,125,126,127,128,129,130,131,132,133,134,135,136,137,138,139,140,141,142,143,144,145,146,147,148,149,150] (prevent the creation of duplicate tracks in OTH distances)
5.	Process maritime image	[53,57,151,152,153,154,155,156,157,158,159,160,161]
6.	Predict vessel trajectory—behavior	[162,163,164,165,166,167,168,169,170,171,172,173,174,175,176,177,178,179,180,181,182,183,184,185,186,187,188,189,190]
7.	Exploit knowledge on historical vessel positioning information	[49,50,51,72,98,101,120,169,171,191,192,193,194,195,196,197,198,199,200,201]
8.	Detect AIS anomalies	[40,41,42,43,45,46,47,48,165,167,202,203,204,205,206,207,208,209,210,211,212,213,214,215,216,217,218,219,220,221]
9.	Detect maritime anomaly	[44,47,52,54,121,164,165,199,205,211,222,223,224,225,226,227,228,229,230,231,232,233,234,235,236,237,238,239,240]
10.	Detect cyberthreat	[55,241,242,243]

### 4.2. Data Sources 

In most cases, the AIS is considered the primary sensor for maritime surveillance. The reason is that the AIS is used for passive detection purposes and the identification of unknown tracks detected by radar. The usage of both radar and AIS information provides the composition of the maritime picture. 

As depicted in Figure 6, the AIS is placed at the center of the map, connected with all the sensors found in the literature. The AIS is separated into two main types, land and satellite AIS. This distinction is essential due to the different possible connections with other sensors. The concurrent fusion of data from both AIS types (satellite and land) is also reported in ref. [100]. Furthermore, the carried-out analysis suggests the representative connections, illustrated in Figure 6. High-frequency surface wave radar (HFSWR) and synthetic aperture radars (SARs) are the most commonly used sensors, especially for over the horizon (OTH) operations. In recent years, lidar radars are also used on board USVs. As presented in Figure 6, data from most of the common radar types are fused with AIS sensors, land or satellite AIS. SAR can be used for maritime surveillance when carried by airborne and satellite means. Data fusion aims mostly to achieve the recognition of the maritime picture of an area of interest and provides the composition in a maritime center.

Moreover, in most merchant shipping applications, the data fusion on board a ship involves only X-band or S-band marine radars and AIS, integrated with the ECDIS (or a similar navigation plotter). In the literature, when referring to radar and AIS fusion on board a ship, like in [39], the type of radar used is not typically mentioned. 

The analysis also indicates solutions using SIGINT equipment to analyze the AIS signal, such as direction finders (DF) for AIS/GNSS spoofing mitigation [244] and localization/positional purposes [245,246]. A key observation stemming from the dataset analysis is that multisensory data fusion is used for AIS anomaly detection to increase the reliability of the recognized maritime picture. In this case, the gap that has been identified is to use AIS, radar, and SIGINT data for the enhancement of maritime cybersecurity and the effective detection of cyberincidents.

### 4.3. Operation 

This section provides insights into how data can be processed to perform specific operations. In other words, the operator defines the method that should be followed to implement the data fusion algorithm for the purposes mentioned in Section 4.1. To do this, one should examine the existing literature, focusing on the implemented algorithms. The focus of this study is the investigation if the algorithms can be used for security purposes, including the detection of cyberincidents. Consequently, authors have identified the application of algorithms and have categorized them into three main types: statistical methods, neural networks, and fuzzy logic, to create three main categories of operations, as follows:

Statistical Methods/Algorithms: Data fusion is implemented using statistical algorithms and/or methods that are used for tracking, kinematic analysis, and probability calculations. For tracking, generic algorithms provide statistical calculations, as mentioned in Table 3. Other algorithms are based on Bayesian fusion, Bernoulli filter, or the joint probabilistic division association (JPDA). An additional implementation uses risk as a threshold for the statistical process [195]. Special applications of Bayesian (BN) fusion have been identified as the dynamic BN [117] and the combination of BN with the sum-product algorithm [109]. Finally, some algorithms are relevant for prediction purposes [247].

For the kinematic analysis, most of the implementations are based on trajectory analysis, like trajectory clustering [171] or similarity [179]. Kalman filters, the Ornstein–Uhlenbeck target motion model, and the Gaussian mixture model are also used for kinematic behavior analysis. For probability calculation, the JPDA method and maximum likelihood [59] are also used.

Neural networks and models: Neural models are implemented for prediction purposes, decision making, and accurate calculations. In this context, belief propagation algorithms [38,56,67] and the Ornstein-Uhlenbeck target motion model [162,203] have been used for decision support operations and prediction purposes. Similarly, for prediction purposes, recurrent neural algorithms have also been utilized.

Fuzzy Logic: Fuzzy logic is also utilized in the context of maritime data fusion. Research works [99,116] implement fuzzy multi-factor logic for correlation purposes.

The combination of data and refinement of information aims to increase the likelihood of detecting a cyberattack against surveillance and navigation systems. Detecting a cyberincident will support decision-making related to response and recovery actions, increasing the possibility for operational continuity.

The mapping of the operations, as illustrated in Figure 7, provides a representative situation of the methods/algorithms used and the relative operations, concluding the following:Most of the algorithms are based on statistical processing. In this category, multiple methods have been used, for tracking purposes, analysis of the kinematic behavior of ships, and probabilistic calculations.For safety and security purposes, the kinematic behavior was analyzed using trajectory analysis among other well-known methods (e.g., Kalman filters, Ornstein-Uhlenbeck). Additionally, research works applied neural networks for the prediction of anomalies or security incidents.Statistical methods, neural networks, and fuzzy logic have been used for the detection of maritime anomalies. It is considered that similar capabilities exist for the real-/near-real-time detection of cyberincidents.To detect cyberincidents, statistical algorithms can be employed to identify anomalies in sensor status or data flow, such as through Bayesian fusion [55], or to recognize network attacks [243]. Additionally, these statistical methods can calculate threshold values or percentages to determine if anomalies are due to a cyberincident. In current applications, the AIS is the sensor typically involved [241]. Furthermore, neural networks are utilized to predict potential cyberthreats and to examine data integrity. If the detection algorithms are integrated into management systems such as the ECDIS, they can enhance the maritime situational awareness, thereby improving the MCSA [256].

### 4.4. Overall Analysis 

Considering the analysis of the relevant manuscripts with the three key elements above (data sources, purpose, and operation), one can conclude the following:Common elements used for MSA, such as radar, AIS [257], and SIGINT data, could be fused to enhance cyberthreat detection capabilities. Thus, sensors used for MSA can also be used for MCSA.Depending on the maritime sensor used, the data expected by the system vary. Such data may include speed, location, course, and other parameters. Then, the correlation of the available maritime data can be used to investigate the vulnerability exploitation of any involved sensor. In situations where the target cannot be detected by radar, which is considered the primary/active sensor, the algorithms should be able to clarify if the target is identified by AIS. For this purpose, statistical calculations can verify the validity and reliability of the data and calculate the cyber risk. Bayesian fusion, Kalman filters, and Gaussian models are only part of the available solutions for such operations, as presented in Section 4.3. When the target cannot be detected by either radar or AIS, the use of direction finding (DF) is suggested to provide signal intelligence (SIGINT) data for further examination of the unknown’s track location. Consequently, the data fusion of radar, AIS, and DF is intended to provide better capabilities to identify/classify the navigational behavior [258].The aim of the fusion algorithms is to detect cyberthreats targeting the main elements used for maritime surveillance and navigation systems, such as AIS, ECDIS, and Radar/ARPA. Such detection processes should be applicable in real time or near-real time, using relevant mechanisms. In addition, the statistical or behavioral analysis of the maritime data (sensors with gray color in Figure 8) can be useful for the long-term and large-scale integration of data, permitting the spatiotemporal analysis to determine and classify a maritime anomaly including cyberthreats [259,260]. For example, possible use cases for each system/sensor include the following:○Detection of AIS spoofing, hijacking, data manipulation, and denial of service (DoS) [261]. Incidents include the sudden change of AIS parameters, AIS spoofing, and AIS transponder on/off as a prior action for a ship to engage in illegal activities, etc.○Detection of malware attacks against ECDIS, causing the subversion of sensor data and misrepresentation and positioning spoofing [261].○Detection of DoS and obfuscation attacks against radar or other electronic warfare attacks such as jamming [34].

## 5. Future Directions

The aim of this survey paper was to examine the existing combination of sensors/systems, including AIS and radar, operating on ships and onshore maritime centers that can be used for detecting cyberattacks. Furthermore, a data fusion approach was presented that can make use of the available maritime sensor information for the purpose of cyberthreat detection. The knowledge gained can drive the design of new threat detection endeavors, focusing on the timely detection of cyberthreats to enhance an organization’s resilience and responsiveness to restore the availability of affected operations and contribute to reliable decision-making [262]. Future investigations will include the detailed presentation of the implementation and evaluation of the algorithm in real/near-real time for the detection of cyberattacks in the integrated bridge systems (maritime OT). Investigations will be supported through a maritime cyber range, an environment that can provide a realistic simulation environment that can be further be utilized for training purposes. The need to enhance the knowledge and skills of IT/non-IT personnel and decision-makers on emerging maritime cyberthreats is crucial [263]. This can be achieved by designing cybersecurity curricula with engaging learning material and activities [264] that are developed over a maritime cyber range, effectively enhancing competencies related to threat detection and response [265]. 

## 6. Conclusions

Cyberattacks in the maritime domain may cause a serious impact on the security of a maritime asset but also may change the safety situation of the maritime asset, especially the ships. In the case of a successful cyberattack against a maritime system/asset, the incident may endanger human lives or impact the global supply chain and economy. This work investigated the fusion of radar, AIS, and SIGINT data and provided insights as to how these data could be transformed and utilized to effectively detect cyberincidents in real/near-real time and achieve not only MSA but also MCSA. Finally, the potential capabilities of data fusion algorithms are determined, for the purpose of detecting cyberincidents, against maritime surveillance and navigation systems.

## Figures and Tables

**Figure 1 sensors-24-03458-f001:**
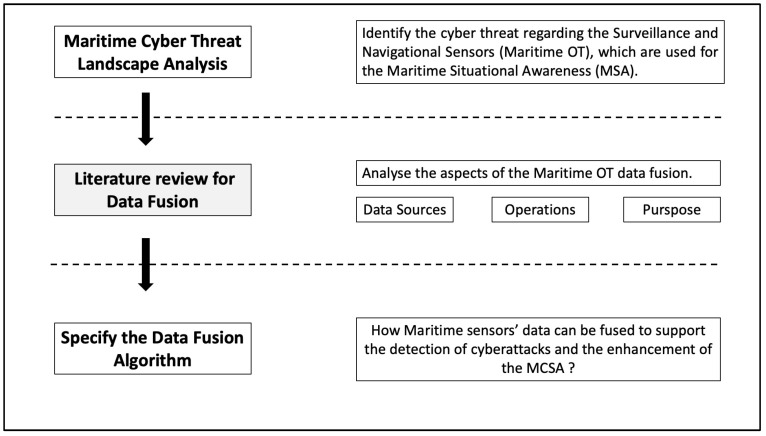
Three-step methodological approach.

**Figure 2 sensors-24-03458-f002:**
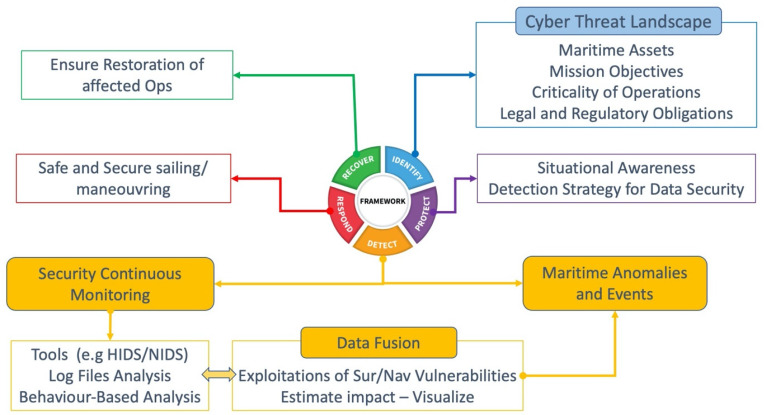
Maritime data fusion across the NIST CFW.

**Figure 3 sensors-24-03458-f003:**
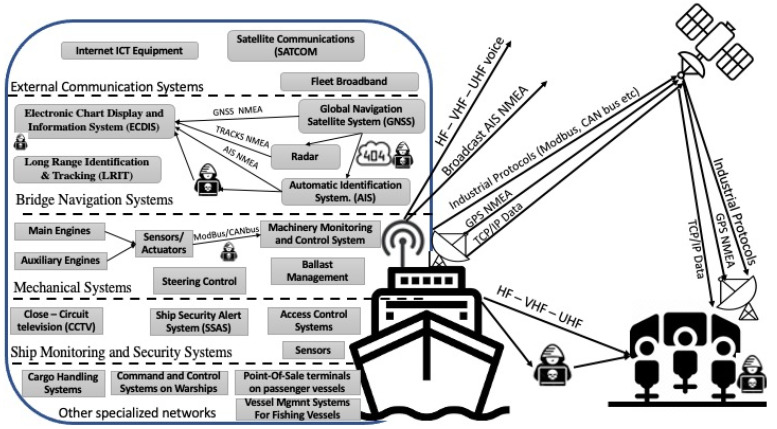
Maritime components in cyberspace.

**Figure 4 sensors-24-03458-f004:**
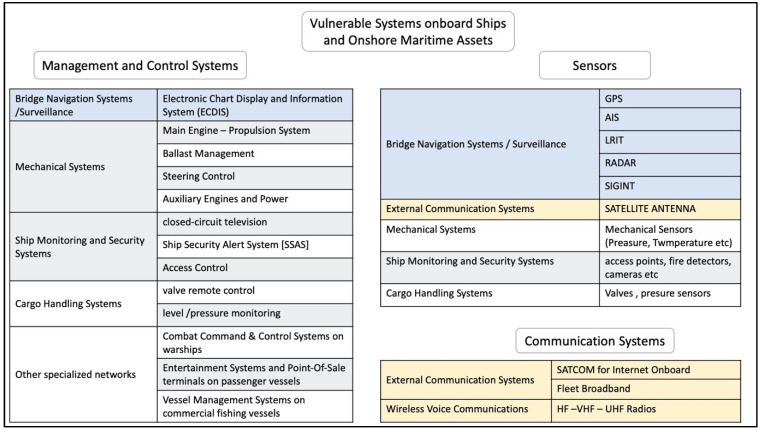
Vulnerable systems in the maritime domain.

**Figure 5 sensors-24-03458-f005:**
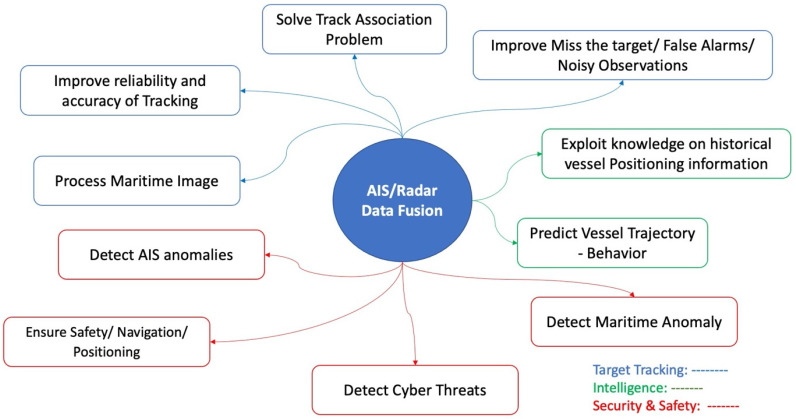
Mapping of radar/AIS data fusion purposes.

**Figure 6 sensors-24-03458-f006:**
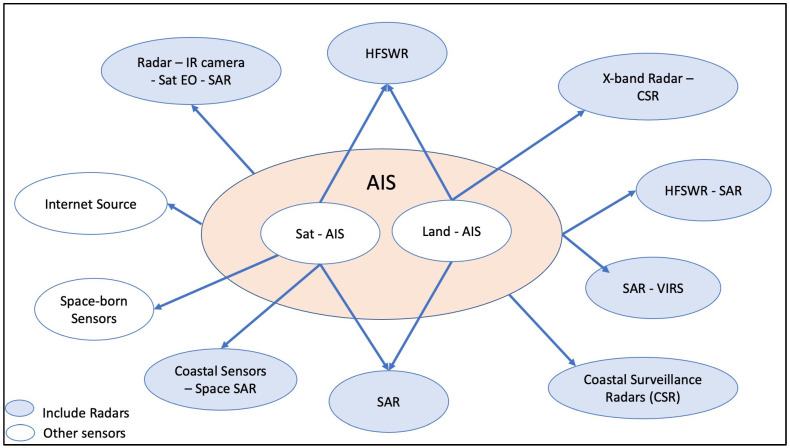
Sensors used for radar/AIS data fusion.

**Figure 7 sensors-24-03458-f007:**
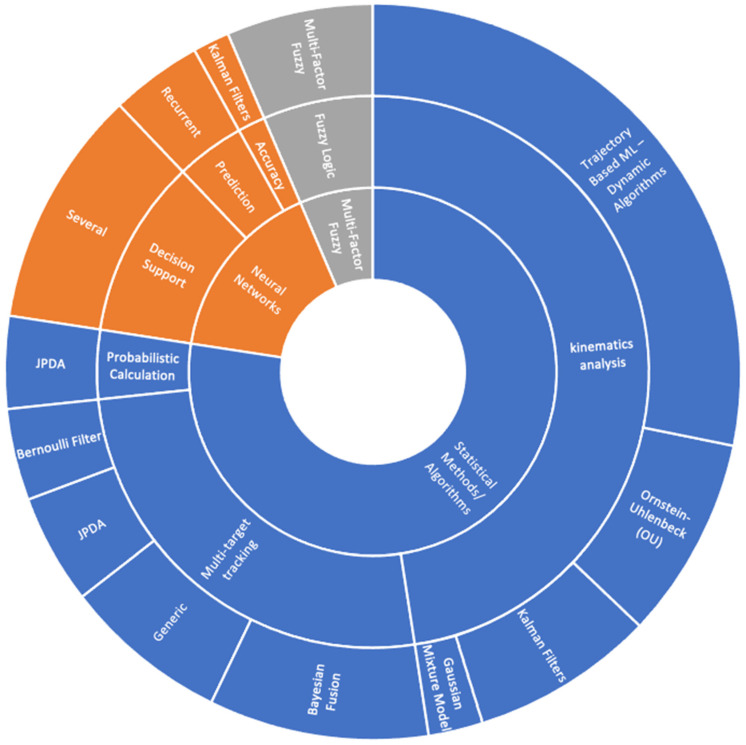
Mapping radar/AIS data fusion operations.

**Figure 8 sensors-24-03458-f008:**
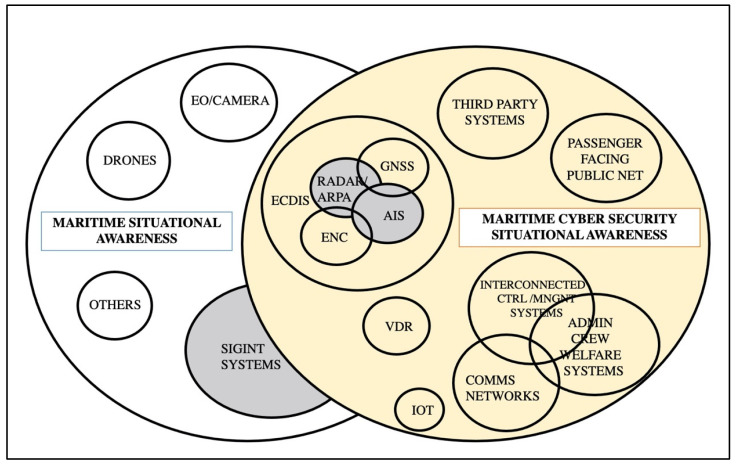
Elements of the Maritime Cyber Situational Awareness.

**Table 1 sensors-24-03458-t001:** Cyberincidents, which involve navigational and surveillance means.

Reported Events/Incidents				
A/A	Incident	Year	Infected System	Type of Attack
1.	NCC group demonstrated ECDIS compromisation, with the use of a portable USB disk by a crew member or through exploitation of an unpatched vulnerability via the internet, after unauthorized access [28].	2015	ENC—ECDIS	Malware/virus
2.	Compromised ECDIS of a warship, which oversaw displaying digital nautical charts. Incident (cyberattack altering nautical maps) reported by US Navy [28].	2015	ENC—ECDIS	-
3.	McAfee found a vulnerability that was exploited through ransomware. This vulnerability allowed us to take total control over propulsion systems and navigation. It was possible to infect the vessel through an unsecured network connection. The attacker was able to encrypt essential system components so no one could control the ship [29,30].	2015	IT/OT Systems	Malware/ ransomware
4.	In this incident, the navigation system of a cargo ship was lost for hours. This incident prevented the captain from controlling the ship’s course between Cyprus and Djibouti [31].	2017	Navigation system	-
5.	A collision happened in Singapore between the US Navy destroyer USS McCain and the small merchant ship Alnic MC. The sources do not tell the whole story, since the US Army was involved. A probable scenario is that Alin MC was attacked by hackers, and then because of that, the collision happened. Ten sailors on board USS died, and five others were injured [31,32].	2017	IT/OT systems	-
6.	GPS spoofing was performed in the Black Sea. Many ships were affected; fortunately, none of the ships were damaged [31].	2017	GNSS	Spoofing
7.	A malware attack on ECDIS infected the Windows system, via a USB stick [31].	2017	ECDIS	Malware
8.	Malware affected MSC, of which the latter incident shut down the ship owner’s Geneva HQ for five days [33].	2020	IT systems	Malware

**Table 3 sensors-24-03458-t003:** Operations for AIS/radar data fusion.

A/A	Main Category	Topic	Algorithm Used	Representative Publications
1.	Statistical methods/ algorithms	Multi-target tracking	Generic	[15,59,65,70,100,112,114,163,205]
2.			Bayesian fusion	[44,56,64,109,117,169,189,238,240,247,248,249]
3.			JPDA	[103,119,122,145,148,223]
4.			Bernoulli filter	[80,122,205,227,250]
5		Kinematics analysis	Kalman filters	[44,72,99,101,106,135,137,145,178,179]
6.			Ornstein-Uhlenbeck (OU)	[40,42,73,84,121,125,173,203,225,227,251]
7.			Trajectory-based ML—dynamic algorithms	[43,49,50,60,74,82,88,97,123,131,138,140,164,171,174,176,179,180,181,182,184,186,188,190,191,197,198,200,201,214,228,231,233,235,252]
8.			Gaussian mixture model	[107,177,196]
9.		Probabilistic Calculation	JPDA	[103,111,119,122,223]
10.	Neural networks	Prediction	Recurrent	[48,175,187,211,253]
11.				
12		Accuracy	Kalman filters	[113,115]
13.		Decision support	Several	[58,68,76,89,92,93,105,108,128,130,161,254,255]
14.	Fuzzy logic	Multi-factor fuzzy		[58,81,104,116,124,126,185,207]

## Data Availability

No new data were created.

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
