# Peer review of "Enhancing Maritime Cybersecurity through Operational Technology Sensor Data Fusion: A Comprehensive Survey and Analysis"

_sensors, 2024, doi:10.3390/s24113458_

Round 1

Reviewer 1 Report

Comments and Suggestions for Authors

The manuscript entitled "Enhancing Maritime Cybersecurity through OT Sensor Data Fusion: A Comprehensive Survey and Analysis" addresses a pivotal area in the realm of maritime security. The subject is intriguing and the manuscript is organized commendably. However, there are several aspects that require attention to elevate the scholarly rigor and relevance of the work to the intended audience.

Primary Concerns:

- The manuscript leans heavily on theoretical analysis and literature surveys. It is essential for the authors to incorporate empirical data or case studies to substantiate the theories proposed. This inclusion would markedly enhance the manuscript’s practical value and scholarly depth.

- While the literature review is extensive, the manuscript must delineate its unique contributions more clearly. It is imperative to articulate how this study advances beyond the existing research in maritime cybersecurity.

Additional Observations:

- The manuscript is well-structured overall; however, the transitions between sections could be smoother to ensure a more coherent narrative flow. Enhancing these transitions would significantly improve the readability and persuasive power of the manuscript.

- The graphical representations in the manuscript could be more informative. It is suggested to increase the size and detail of the figures to aid in comprehension. Additionally, a more thorough explanation accompanying each figure would be beneficial.

- The manuscript should undergo meticulous revision to remove redundant content that overlaps significantly with prior publications by the authors. It is crucial to compare and emphasize the novel aspects of this study clearly.

- The explanation of key technical terms and methodologies, particularly those pertaining to data fusion techniques in maritime environments, is somewhat lacking. Providing a more detailed explanation of these terms would aid in the manuscript’s accessibility to a broader audience.

- Guidance on how readers should interpret the results and implications of the study is necessary. It would be advantageous to elucidate the practical applications of the research findings in enhancing maritime cybersecurity.

- Please verify the consistency of technical notation and terminology throughout the manuscript. For instance, the usage of specific terms related to sensor data fusion needs uniformity to avoid confusion.

Comments on the Quality of English Language

The manuscript's command of the English language is adequate for an academic audience, but there are areas where improvement is necessary to meet the publication standards of the journal. Here are specific points that need attention:

- The manuscript exhibits occasional grammatical errors that could hinder comprehension and detract from the professional quality of the writing. Rigorous proofreading and correction of these errors are advised to maintain the integrity of the academic discourse.

- Some sentences are overly complex or awkwardly constructed, which can challenge reader comprehension, especially for those for whom English is not a first language. Simplifying these constructions will enhance the readability and ensure that the scientific content is accessible to a wider audience.

- There are instances where technical terms and key concepts are not used consistently throughout the manuscript. Ensuring uniformity in terminology is crucial for clarity and to avoid potential confusion among readers.

- The manuscript should maintain consistent verb tenses throughout. Fluctuations in tense can disrupt the narrative flow and make the text appear less polished and coherent.

- The manuscript occasionally suffers from verbosity and redundancy. Streamlining the content to eliminate unnecessary repetition would not only reduce the length but also improve the impact and clarity of the information presented.

Author Response

Review Report (Reviewer 1)

Primary Concerns:

- The manuscript leans heavily on theoretical analysis and literature surveys. It is essential for the authors to incorporate empirical data or case studies to substantiate the theories proposed. This inclusion would markedly enhance the manuscript’s practical value and scholarly depth.

  • A new Section 4.4 summarizes the analysis outcomes based on three key elements: data sources, purpose, and operations. This section now includes case studies focusing on developing cyber threat detection capabilities for ECDIS, AIS, and Radar, thereby enhancing the manuscript's practical value.

- While the literature review is extensive, the manuscript must delineate its unique contributions more clearly. It is imperative to articulate how this study advances beyond the existing research in maritime cybersecurity.

To highlight the study's unique contributions, we have made several modifications:

  • A clear statement of contributions in the abstract.
  • Definition of scope in the introduction.
  • A new paragraph in Section 4 linking the environment identification with literature review findings.
  • Renaming of Subsection 3.2.1 and Section 3 to clarify the study's scope.

Additional Observations:

- The manuscript is well-structured overall; however, the transitions between sections could be smoother to ensure a more coherent narrative flow. Enhancing these transitions would significantly improve the readability and persuasive power of the manuscript.

  • We have refined the transitions at the beginning of Section 3, the end of Section 3, and the start of Section 4. Clear findings have been added to Section 4 for better readability.
  • Transition between section 4 and section 5 has been added.
  • The description of the methodology is completely changed, to describe effectively the flow of the study, especially in section 3.

- The graphical representations in the manuscript could be more informative. It is suggested to increase the size and detail of the figures to aid in comprehension. Additionally, a more thorough explanation accompanying each figure would be beneficial.

  • The size of all figures has been increased, with significant modifications to Figure 6 to clarify the results.

- The manuscript should undergo meticulous revision to remove redundant content that overlaps significantly with prior publications by the authors. It is crucial to compare and emphasize the novel aspects of this study clearly.

  • Redundant content has been eliminated, and previous Section 5 was removed to prevent overlap. Key contents from the former Section 5 have been integrated into Section 4 to underscore the novel findings.

- The explanation of key technical terms and methodologies, particularly those pertaining to data fusion techniques in maritime environments, is somewhat lacking. Providing a more detailed explanation of these terms would aid in the manuscript’s accessibility to a broader audience.

  • We have clarified technical terms and methodologies wherever possible, and some sentences have been rephrased to be more comprehensible to a wider audience.

- Guidance on how readers should interpret the results and implications of the study is necessary. It would be advantageous to elucidate the practical applications of the research findings in enhancing maritime cybersecurity.

  • Better explanation has been added to section 3 to describe the connection between the 3 phases. Section 4.4 has been added to describe better the findings and conclude with a third step to define the necessity of developing data fusion to support MCSA
  • Additional description has been done to the section 5. Future direction connected previous authors work with the future investigation.

- Please verify the consistency of technical notation and terminology throughout the manuscript. For instance, the usage of specific terms related to sensor data fusion needs uniformity to avoid confusion.

  • Confusing terms have been rephrased or removed to ensure clarity.

Comments on the Quality of English Language

The manuscript's command of the English language is adequate for an academic audience, but there are areas where improvement is necessary to meet the publication standards of the journal. Here are specific points that need attention:

- The manuscript exhibits occasional grammatical errors that could hinder comprehension and detract from the professional quality of the writing. Rigorous proofreading and correction of these errors are advised to maintain the integrity of the academic discourse.

  • Comprehensive proofreading and correction of grammatical errors have been completed.

- Some sentences are overly complex or awkwardly constructed, which can challenge reader comprehension, especially for those for whom English is not a first language. Simplifying these constructions will enhance the readability and ensure that the scientific content is accessible to a wider audience.

  • Sentences on pages 2, 4, 5, 10, 16, 17, and 18 have been rephrased to improve readability and accessibility.

- There are instances where technical terms and key concepts are not used consistently throughout the manuscript. Ensuring uniformity in terminology is crucial for clarity and to avoid potential confusion among readers.

  • We have carefully reviewed and adjusted verb tenses for consistency.

- The manuscript should maintain consistent verb tenses throughout. Fluctuations in tense can disrupt the narrative flow and make the text appear less polished and coherent.

  • The maintainance of consistent verb tenses have taken into consideration.

- The manuscript occasionally suffers from verbosity and redundancy. Streamlining the content to eliminate unnecessary repetition would not only reduce the length but also improve the impact and clarity of the information presented.

  • Redundant content and repetitions have been reduced. Removing Section 5 and restructuring Section 4 proved beneficial in this regard.

Reviewer 2 Report

Comments and Suggestions for Authors

Cyber security is becoming an increasingly important aspect in ensuring maritime data protection and operational continuity. The authors submit a review manuscript to sensors about “Enhancing Maritime Cybersecurity through OT Sensor Data Fusion: A Comprehensive Survey and Analysis”. They conducted a survey investigating whether the fusion of OT data may also improve the ability to detect cyber incidents in real time or near real time. Thorough analysis and data fusion approaches are included. The following issues should be considered in the revision of the manuscript:

1. The authors discuss the potential of AIS and radar data fusion for detecting anomalies and cyber threats in maritime environments, highlighting the importance of real-time detection capabilities. The authors can include a comparative analysis of the performance of AIS and radar data fusion with other sensor fusion techniques, such as SAR or HFSWR, to provide a more comprehensive understanding of the state-of-the-art in maritime cybersecurity.

2. This article presents a comprehensive survey and analysis on enhancing maritime cybersecurity through the fusion of OT sensor data, with a focus on the role of navigation and surveillance systems in improving the MCSA. It would benefit from a more detailed discussion on quantum information aspects, particularly in the context of sensor data encryption [Phys. Rev. Lett. 130, 250801 (2023); Nature 501, 69 (2013)] and secure communication [Science Advances 10, eadk3258 (2024); Light: Science & Applications 12, 175 (2023)], which are critical for the MCSA.

3. The author needs to check the full text carefully for spelling errors and grammar issues.

Comments on the Quality of English Language

Minor editing of English language required

Author Response

Review Report (Reviewer 2)

  1. The authors discuss the potential of AIS and radar data fusion for detecting anomalies and cyber threats in maritime environments, highlighting the importance of real-time detection capabilities. The authors can include a comparative analysis of the performance of AIS and radar data fusion with other sensor fusion techniques, such as SAR or HFSWR, to provide a more comprehensive understanding of the state-of-the-art in maritime cybersecurity.
  • Section 4 restructured to describe in a more efficient way the findings of the survey. The vulnerabilities and use cases related with the AIS has been identified.
  • Use cases include the potential impact of the cyber threats against radars (SAR or HFSWR).
  1. This article presents a comprehensive survey and analysis on enhancing maritime cybersecurity through the fusion of OT sensor data, with a focus on the role of navigation and surveillance systems in improving the MCSA. It would benefit from a more detailed discussion on quantum information aspects, particularly in the context of sensor data encryption [Phys. Rev. Lett. 130, 250801 (2023); Nature 501, 69 (2013)] and secure communication [Science Advances 10, eadk3258 (2024);Light: Science & Applications 12, 175 (2023)], which are critical for the MCSA.
  • Information aspects, including sensor data encryption has been added in section 4 completing the findings.
  1. The author needs to check the full text carefully for spelling errors and grammar issues.
  • Comprehensive proofreading and correction of grammatical errors have been completed.

Round 2

Reviewer 2 Report

Comments and Suggestions for Authors

Because the revised manuscript does not indicate where the changes were made, it is very difficult to read. It is necessary for the authors to summarize the work related to security algorithms carefully.

Comments on the Quality of English Language

Minor editing of English language required

Author Response

1. Changes from the 1st and 2nd rounds of the review have been highlighted. 

2. A paragraph summarizing the work related to security algorithms has been added.

3. Minor changes have been made throughout the document, to improve/correct English language.